# In Utero Exposure to Persistent Organic Pollutants and Childhood Lipid Levels

**DOI:** 10.3390/metabo11100657

**Published:** 2021-09-28

**Authors:** Maegan E. Boutot, Brian W. Whitcomb, Nadia Abdelouahab, Andrea A. Baccarelli, Amélie Boivin, Artuela Caku, Virginie Gillet, Guillaume Martinez, Jean-Charles Pasquier, Jiping Zhu, Larissa Takser, Lindsay St-Cyr, Alexander Suvorov

**Affiliations:** 1Department of Biostatistics and Epidemiology, School of Public Health and Health Sciences, University of Massachusetts Amherst, Amherst, MA 01003, USA; mboutot@umass.edu (M.E.B.); bwhitcomb@schoolph.umass.edu (B.W.W.); 2Department of Obstetrics and Gynecology, Faculty of Medicine and Health Sciences, Sherbrooke University, Sherbrooke, QC J1H 5N4, Canada; nadia.abdelouahab@usherbrooke.ca (N.A.); jean-charles.pasquier@USherbrooke.ca (J.-C.P.); 3Department of Environmental Health Sciences, Mailman School of Public Health, Columbia University, New York, NY 10032, USA; andrea.baccarelli@columbia.edu; 4Department of Pediatrics, Faculty of Medicine and Health Sciences, Sherbrooke University, Sherbrooke, QC J1H 5N4, Canada; amelie.boivin@usherbrooke.ca (A.B.); virginie.gillet@usherbrooke.ca (V.G.); lindsay.st-cyr@usherbrooke.ca (L.S.-C.); 5Department of Biochemistry and Functional Genomics, Faculty of Medicine and Health Sciences, Sherbrooke University, Sherbrooke, QC J1H 5N4, Canada; artuela.s.caku@usherbrooke.ca; 6Department of Chemistry, Faculty of Sciences, Sherbrooke, QC J1K 2R1, Canada; guillaume.martinez@usherbrooke.ca; 7Environmental Health Science and Research Bureau, Health Canada, Ottawa, ON K1A 0K9, Canada; jiping.zhu@canada.ca; 8Department of Pediatrics & Department of Psychiatry, Faculty of Medicine and Health Sciences, Sherbrooke University, Sherbrooke, QC J1H 5N4, Canada; larissa.takser@usherbrooke.ca; 9Department of Environmental Health Sciences, School of Public Health and Health Sciences, University of Massachusetts Amherst, Amherst, MA 01003, USA

**Keywords:** dyslipidemia, fetal programing, lipid metabolism, NAFLD, persistent organic pollutants

## Abstract

Animal studies have shown that developmental exposures to polybrominated diphenyl ethers (PBDE) permanently affect blood/liver balance of lipids. No human study has evaluated associations between in utero exposures to persistent organic pollutants (POPs) and later life lipid metabolism. In this pilot, maternal plasma levels of PBDEs (BDE-47, BDE-99, BDE-100, and BDE-153) and polychlorinated biphenyls (PCB-138, PCB-153, and PCB-180) were determined at delivery in participants of GESTation and Environment (GESTE) cohort. Total cholesterol (TCh), triglycerides (TG), low- and high-density lipoproteins (LDL-C and HDL-C), total lipids (TL), and PBDEs were determined in serum of 147 children at ages 6–7. General linear regression was used to estimate the relationship between maternal POPs and child lipid levels with adjustment for potential confounders, and adjustment for childhood POPs. In utero BDE-99 was associated with lower childhood levels of TG (*p* = 0.003), and non-significantly with HDL-C (*p* = 0.06) and TL (*p* = 0.07). Maternal PCB-138 was associated with lower childhood levels of TG (*p* = 0.04), LDL-C (*p* = 0.04), and TL (*p* = 0.02). Our data indicate that in utero exposures to POPs may be associated with long lasting decrease in circulating lipids in children, suggesting increased lipid accumulation in the liver, a mechanism involved in NAFLD development, consistent with previously reported animal data.

## 1. Introduction

Polybrominated diphenyl ethers (PBDEs) are related brominated organic chemicals used widely as flame-retardants in synthetic materials, including clothing. Concentrations of PBDEs have increased in a broad range of environmental samples and human media over approximately 30 years [1]. Despite withdrawal from the US market by 2013, current human exposure to PBDEs remains close to their historical maximum in North America due to their presence in consumer products and other materials in households and other living locations. Some studies suggest that human PBDE burdens may even continue to increase because of the persistence of PBDEs in the environment [2]. Because of their lipophilicity and chemical stability [3,4] PBDEs accumulate in adipose tissue and are mobilized in pregnancy [5]. As a result, the developing fetus is exposed via cord blood [5]. These findings are supported by the fact that PBDEs are found in the majority of fetal samples in North America [6,7].

In the liver, there is an overlap between lipid metabolism and detoxification pathways of drugs and xenobiotics. For example, xenobiotic-inducible nuclear receptors regulate both xenobiotic metabolism and lipid homeostasis [8,9,10,11,12,13,14,15,16,17,18]. In addition, expression of hepatic enzymes involved in metabolism of xenobiotics and lipids differ significantly by sex [19,20,21,22,23,24]. In humans, 1249 sex-specific genes have been identified in the liver [22], and found to be enriched for lipid metabolism and cardiovascular disease among other biological categories. These data suggest induction of lipid metabolism pathways after chemical exposures and sexually dimorphic character of this response. 

Concerningly, PBDEs exposure produce transient effects on lipid metabolism via reversible induction of liver nuclear receptors, constitutive androstane receptor (CAR) and pregnane X receptor (PXR) [25]. Recent rodent studies demonstrate however, that developmental exposures to PBDEs may result in permanent effects on lipid metabolism leading to dyslipidemia and a non-alcoholic fatty liver disease (NAFLD)-like phenotype [26,27]. The molecular basis of these programing events is not well understood but likely includes liver epigenome remodeling [28].

In this study, based on our results from animal studies [26,29], we hypothesized that in utero exposures to PBDEs would produce long lasting effects in liver metabolism of lipids in humans. To test this hypothesis, we evaluated the associations between maternal blood organohalogens at delivery and lipid profiles of children at 6–7 years of age in the prospective GESTation and Environment (GESTE) cohort.

## 2. Results

Our analysis of dyads with complete data on in utero exposure, age 6–7 childhood lipids, and covariates represents substantial dropout in this longitudinal study among women who provided blood samples and information at delivery. Comparisons of those included in analysis (*n* = 147, 25%) with those who excluded due to dropout (*n* = 413, 75%) identified small differences in age (29.3 vs. 28.4, *p* = 0.04) and BDE-153 (0.012 vs. 0.022, *p* = 0.03), but no differences were observed in other in utero chemical concentrations or BMI. In addition, at baseline, only a subset of women chose to participate in follow-up visits, further suggesting randomness to dropout and low probability of selection bias as a result.

Sample summary statistics are presented in Table 1 and Table 2. Table 1 displays participant-pair (*n* = 147) characteristics and serum levels for selected metals by tertiles of child’s total lipid levels. Maternal age significantly (*p* < 0.05) varied by total lipid levels. Manganese levels at delivery also varied significantly by total lipids; however, this variable was not associated with any of the individual PCBs, PBDEs, or the PCB or PBDE summation variables (range of t-test *p*-values: 0.53–0.996, data not shown). Based on bivariate analyses, maternal age at pregnancy and post-pregnancy BMI were included as potential confounders in adjusted, multivariable models.

Overall, most POPs were not strongly associated with childhood lipid levels in simple or adjusted models; however, PCB-138 and BDE-99 were moderately (*p* < 0.10) or significantly (*p* < 0.05) associated with multiple lipid measurements (Table 3). PCB-138 was associated significantly with TG (*p* = 0.04), LDL-C (*p* = 0.04) and TL (*p* = 0.02) and associated moderately with TCh (*p* = 0.09) and HDL-C (*p* = 0.07). BDE-99 was significantly associated with TG (*p* = 0.003) and moderately associated with HDL-C (*p* = 0.06) and TL (*p* = 0.07) (Table 3). Additionally, PCB-180 was significantly associated with TG (*p* = 0.03) and sum of PCB was moderately associated with TL (*p* = 0.09). Associations between HDL-C and PCB-138 or BDE-99 were positive, while all other moderate and significant associations between lipid parameters and POPs were negative. Tertile analyses supported analyzing associations as linear relationships (data not shown).

We conducted analysis to address potential confounding by childhood chemical levels, which could result in a misattribution of effects of chemical exposure in childhood on lipids as being due to in utero exposure. Because of a primary goal to assess and control for confounding, we first evaluated correlations between in utero chemical levels and those measured in children age 6–7 (Appendix A) and observed mostly weak relations between maternal chemical levels and those among children at age 6–7. Levels of childhood BDE-153 were correlated with maternal delivery values for BDE-100 (r = 0.24, *p* =0.005) and BDE-153 (0.34, *p* <0.0001) (*n* = 140 child-mother pairs). We ran main analyses adjusted for BDE-153 to estimate in utero effects independent of childhood chemical levels (Appendix A) (*n* = 136 child-mother pairs). Generally, results remained the same or were strengthened. We also conducted analysis to evaluate the relationship between age 6–7 years old PBDE exposures with serum lipids (Appendix A). For these analyses, we utilized wet weight chemicals in order to avoid adjusting exposure for the dependent variable and resulting bias along with interpretational challenges. Associations between childhood levels of BDE-47 and 99 were observed with lower HDL-C and higher LDL-C. BDE-100 was also associated with lower HDL-C. 

Analyses stratified by child sex suggested sex-specific associations (Figure 1). In the majority of stratified analyses, the associations among girls and boys were not equivalent to one another nor the non-stratified (“All”) results and followed a similar trend. Nevertheless, tests for interactions between chemical and child sex were not significant in bivariate or adjusted models (Table 4).

## 3. Discussion

Several rodent studies observed long-term programing of lipid metabolism by in utero exposures to POPs [26,27,29,30,31]. Results from the current study of mother-child pairs in the GESTE cohort suggest that these previously reported findings may apply for humans, as well. In mice, we have found that in utero exposure to PBDEs causes a shift of lipid balance between the liver and blood, whereby increased uptake and accumulation of lipids by the liver results in decreased TG in blood and a NAFLD-like liver phenotype. In our current study, we also observe decreased TG in blood associated with in utero exposures to BDE-99 and PCB-138. Additionally, in utero levels of BDE-99 were associated with lower total lipids, and those of PCB-138 were associated with lower LDL-C and total blood lipids in 6–7-year-old children. These associations between in utero POPs and lower plasma lipid levels are consistent with increased uptake of fatty acids by the liver and may indicate higher fat accumulation in children’s livers. 

Notably, PCB-138 and BDE-99 had the broadest normalized ranges of exposures among all compounds analyzed, suggesting that weaker associations between other POPs at delivery and lipid profiles at childhood may be due to our limited ability to detect them and/or distribution in the study sample, rather than due to the lack of biological link between exposures and outcomes. In addition, adjustment for POPs measured in childhood plasma samples did not weaken estimates of the in utero exposure effects as represented by maternal plasma levels. The potential for childhood POPs to confound the relationship of in utero exposure with childhood lipid levels depends upon common causes that result in correlation between exposure at these two time points. We did not observe such a correlation in our data, suggesting no substantial confounding of the relationship of interest. This provides further support that association between in utero POPs are due to programing effects rather than due to continuous exposures lasting into childhood, though caution is warranted to acknowledge the potential for chance findings

Further, we observed indication that programing is sexually dimorphic and that the relationship between in utero POPs and lipid profile in children is sex specific. Because our analysis was inherently restricted to dyads with available data on both in utero chemical levels and childhood lipid levels, our sample size was modest and we had limited statistical power to detect interactions. In analyses stratified on offspring sex, we observed consistent differences in the associations of in utero POPs and childhood lipids. This observation is in agreement with our hypothesis that sex specificity in liver metabolic functions results in modified response of lipid liver metabolism to chemical exposures. Our limited ability to find significant effect of sex may be also explained by the young age of the study participants. Liver sex specificity is generally thought to be established after the onset of puberty, when growth hormone secretion reaches clear sexually dimorphic patterns [32]. Resent research has demonstrated however, that even late gestation fetal livers have well-pronounced sex-specific pattern of gene expression in mice and more than 70% of fetal sex-biased genes overlap with adult sex-biased genes [33]. Comparable data are not available for humans to our knowledge. 

To the best of our knowledge, this study is the first epidemiologic investigation of a putative link between in utero exposures to PCBs and PBDEs and long-term changes in serum lipids in humans. Exposure, levels to PBDEs in the GESTE cohort are comparable with exposure levels in many studies in the North America as compared elsewhere [34,35]. Maternal plasma levels of PCBs were very close to these in a representative sampling of the U.S. population in the National Health and Examination Survey [36]. Effects of developmental exposures to PBDE on permanent changes in liver lipid metabolism and circulating lipids have been shown in several rodent studies [26,27,29,30,31]. For example, mice exposed to low doses of BDE-47 perinatally and then kept on a high fat diet for 14 weeks developed dyslipidemia and hepatic steatosis [27]. A significant body of literature exists linking adult exposure to PCBs with altered lipid metabolism, including NAFLD promotion [37]. We were unable to identify studies in which long lasting changes in lipid metabolism outcomes were analyzed following developmental exposures to PCB. However, in four-month-old rats exposed perinatally to non-dioxin-like PCBs changes in gene expression in blood were significantly enriched for lipid metabolism [38].

Transient activation of liver Phase I-III enzymes via induction of nuclear receptors (xenosensors) by a broad range of xenobiotics is one of the central concepts in toxicology. Given that this same set of nuclear receptors regulates lipid homeostasis [8,9,10,11,12,13,14,15,16,17,18], potential short-term responses to xenobiotics include transient changes in liver lipid metabolism. PBDEs and PCBs are no exception to this rule, as both groups of chemicals are ligands for CAR and PXR xenosensors [25]. In other words, acute effects of PBDE and PCB on lipid levels are predictable and represent a common toxicological response. Positive associations of childhood LDL-C with childhood levels of BDE 47 and 99, as well as negative associations of childhood HDL-C with childhood levels of BDE 47, 99 and 100 correspond to this type of response. These results indicate that ongoing exposures to PBDE may represent a risk factor for cardiovascular disease by shifting the lipid balance toward an increase in “bad cholesterol” and decrease in “good cholesterol” levels. 

The major focus of the current study is of a different nature however, as it relates to long-term programing by in utero exposures. Clinically, this is of significance because chronic health conditions arising from in utero programing affect health throughout the lifespan. Molecular mechanisms involved in this programing are not well understood. Our recent studies in rodents suggest that transient activation of the central metabolic master switch, the mTOR pathway [39], in the liver by developmental exposures to PBDE may result in permanent reprograming of the liver lipid metabolism [26,29] involving changes in the profile of hepatic small non-coding RNA and DNA methylation [28]. It is not clear, however if mTOR activation is mediated via CAR and/or PXR signaling. Understanding of the initial steps triggering liver metabolism reprograming is complicated by the presence of several transcript variants of CAR [40] and PXR [41], which may have different affinity for different PCB and PBDE congeners; their different downstream mode of action [42]; and emerging evidence of mechanistic links between these receptors and the mTOR pathway [42,43]. Additionally, effects of lipid metabolism reprograming by POPs may have different effects in males and females due to the sex-specific differences in liver expression of many lipid metabolism genes [22]. 

Higher lipid uptake by the liver due to long term effects of in utero exposure to POPs, as we report for some chemicals in this pilot study, may contribute to pediatric NAFLD, the most common chronic liver disease in American children [44,45,46] with an estimated prevalence of 13% [45]. Emerging data demonstrates that pediatric NAFLD may be a more aggressive form of liver disease than adult NAFLD [47]. For example, in a recent study of children with NAFLD, one third of participants developed *nonalcoholic steatohepatitis* (NASH) or worsening of fibrosis in a 2-year period [48]. Severely obese adolescents have significantly higher NASH incidence and liver fibrosis then adults [49]. Pediatric NAFLD is associated with greater liver-related morbidity and mortality in adulthood [50]. Identifying potentially modifiable factors that contribute to these outcomes, and development of future interventions, is thus of great public health significance.

Some limitations and cautions are important to note. In our analysis, we expressed PCB and PBDE concentrations per blood volume without adjustment for blood lipids. For our study of in utero POPs, fetal exposure results from total POPs delivered to placenta via maternal circulation regardless of maternal body burden. The lipophilic nature of PCBs and PBDEs presents a challenge for environmental epidemiologic research, and thoughtful consideration of the underlying biology is a critical step to guide statistical analysis [51]. Lipid normalization is not appropriate when circulating chemical levels, rather than total body burden, is the biologically relevant tissue [51]. Additionally, lipid profile itself may be affected by exposures to POPs [26,29,52,53,54]. For statistical modeling of outcomes affected by lipids, or lipids themselves, adjustment or use of lipid standardization results in bias to the parameter of interest [51]. Finally, in blood, many POPs are bound to blood proteins. For example, PBDEs and their metabolites have high affinity to transthyretin in humans [55,56,57] and to major urinary protein in mice [58]. In that case, adjustment for lipids is not relevant for any biologically meaningful estimation of exposure. 

Uncontrolled confounding is always a potential source of bias for observational research. Nevertheless, we were able to adjust for factors including child chemical levels and other variables from the rich GESTE dataset that might represent confounders. Reassuringly, adjustment for covariates had minimal effect on estimates. Thus, through some variables play roles in the relationship of in utero exposure and child lipid levels that are uncertain, such as adiposity which may be a confounder or causal intermediate, results were not impacted. *A priori* power calculations for this exploratory study suggested adequate sample size for the study but did not explicitly address the multiple comparisons due to the number of chemical exposures and lipid outcomes. Results of this exploratory study should be interpreted with caution, accordingly. 

One limitation of this study is that lipids were analyzed in non-fasting blood samples collected from children. As a result, child lipid levels are subject to post-prandial variation. Recent studies suggest however, that non-fasting total cholesterol, HDL-C and LDL-C change minimally in response to food intake and may be superior to fasting levels in predicting adverse health outcomes [59,60,61,62,63]. Specifically, a cross-sectional study of more than 200,000 lipid profiles [64] observed that fasting times account for less than 2% variation for total cholesterol and HDL-C and less than 20% for TG. Given that humans are rarely present in fasting state, non-fasting lipids may better present individual molecular reactions of the organism for lipid intake. As such, non-fasting lipid profiles are becoming more popular as markers of health conditions in clinical guidelines [65,66,67,68]. Thus, using non-fasting lipids in our study is a moderate limitation as it represents one clinically relevant way to assess lipid profiles. No systematic error is anticipated from this procedure as soon as all within study comparisons are based on non-fasting lipids.

## 4. Materials and Methods

### 4.1. Participants

We conducted analyses using data from the prospective GESTE cohort, which is described in detail elsewhere [34,35,69,70]. In short, GESTE cohort is comprised of (a) women residing in the Eastern Townships (Estrie) of Quebec, Canada having a pregnancy between 2007 and 2009 and (b) children born from these pregnancies. Women were recruited at their first prenatal visit by 20 weeks gestational age (*n* = 400, at >20 weeks), or at delivery (*n* = 400 additional women) at the Research Center of the Centre Hospitalier Universitaire de Sherbrooke (CHUS). Eligible criteria included women ≥18 years old at their first prenatal visit without thyroid diseases, who were planning on residing in the Estrie for at least three years. At delivery, only healthy, uncomplicated term pregnancies were included in the cohort.

Our study population was restricted to maternal-child dyads including women with singleton pregnancies, having complete data for maternal PBDEs at delivery as well as a lipid profile performed at age 6–7 years old. After applying these inclusion criteria, a total of 147 observations were included for analysis. 

The Institutional Review Boards of the CHUS, Université de Sherbrooke (IRB # 05-057-S1, 2008-103), Health Canada (REB 2017-035H) and the University of Massachusetts, Amherst (IRB # 2019-5622), approved study protocols, and women provided informed consent for themselves and their children.

### 4.2. Data Collection

Data were collected and managed using the secure online platform REDCap [71]. Non-fasting maternal blood was collected at delivery in 10-mL Vacutainer^®^ Hemogard tubes with ethylenediaminetetraacetic acid (ref. 366643, Becton-Dickinson, San Jose, CA, USA). The plasma was frozen at −20 °C in decontaminated Supelco glass storage tubes (Supelco, Inc., Bellefonte, PA, USA) for PBDEs and PCBs. In addition, 10-mL metal-free blood collection tubes with 0.05 mL of 15% ethylenediaminetetraacetic acid K3 stored at 4 °C were used for manganese, lead, mercury and cadmium analysis. At the age of 6–7 years old, ten mL of non-fasting whole blood were taken from each child using a BD Vacutainer^®^ tubes (ref. 367962, Becton Dickinson, Franklin Lakes, NY, USA) containing lithium-heparin for the analysis of PBDEs and lipid profiles. Non fasting blood samples were stored at −80 °C and analyzed for lipid profiles in the core laboratory of the CHUS. 

### 4.3. Exposure Assessment–PBDEs and PCBs in Utero and at 6–7 

Maternal plasma levels of PBDEs (BDE-47, BDE-99, BDE-100, and BDE-153) and PCBs (PCB-138, PCB-153, and PCB-180) were analyzed in the laboratory of L. Takser (Université de Sherbrooke) using the method described by Covaci and Voorspoels [72]. As previously described, [34,35], analytes were quantified using Varian 4000 Ion Trap GC/MS-MS (gas chromatograph/tandem mass spectrometer) (Varian Inc., CA, USA). Quality control was conducted through regular analyses of water blanks, solvent blanks and random duplicate samples. All blanks were subtracted from sample values on a batch basis. Inter-batch coefficients of variation were ≤10% and ≤8% for PBDEs and PCBs, respectively. Limits of detection (LOD), defined as 3 times the noise level, were established at 0.1 pg/μL for PBDEs and at 0.02 pg/μL for PCBs. Routine checks of accuracy and precision were also performed using reference materials (1589a) from the National Institute of Standards and Technology (Gaithersburg, MD, USA). Recoveries ranged between 75 and 125%. 

Child plasma PBDE levels were analyzed in the laboratory of J. Zhu (Health Canada). Analytical procedures are described in detail elsewhere [69]. In short, analytes were quantified using an Agilent 7890A GC coupled with 7200 QTOF MS (Agilent Technologies, Santa Clara, CA, USA). Quality control was conducted through regular analyses of water blanks, commercial human serum (S7023, Sigma Aldrich, MO, USA) that was spiked with target analytes, and random duplicate samples. All blanks were subtracted from sample values on a batch basis. Inter-batch coefficients of variation were in a range between 1% and 9% for different PBDE congeners. Compound specific limit of detection (LOD) was estimated to be in the range of 1–5 pg/g wet weight. Mean recoveries were in the range of 94–125%, except for BDE-183 (59.6%). The concentrations of target analytes in the samples were recovery-corrected against BDE-138, a congener that was not present in the environment and humans, as the surrogate standard. 

### 4.4. Outcome Assessment–Lipid Profiles in Children at Age 6–7

The lipid work-up was performed in the the biochemistry laboratory of the CHUS. Fresh blood samples were centrifuged at 1300–2000 g for 10 min to separate the plasma from the whole blood within 1 h after collection. Samples were stored at −80 °C. Total cholesterol (TCh) (assay ref. 5168538), total triglycerides (TG) (assay ref. 5171407), and high density lipoproteins (HDL-C) (assay ref. 5168805) were analyzed using enzymatic colorimetric method using a Roche Cobas 8000 analyzer with commercial c702 kit, following manufacturer’s instructions (Roche Diagnostics, Germany). The assay was calibrated for each new reagent bottle or batch or when required by internal control. Analytical variation and systematic error were both below 3% [69]. Lower limits of detection (LOD) were 0.1 mmol/L (3.86 mg/dL) for TCh, 0.08 mmol/L (3 mg/dL) for HDL-C and 0.1 mmol/L (8.85 mg/dL) for TG. Concentration of low-density lipoproteins (LDL-C) was estimated using the Friedewald formula (LDL = TCh − HDL − TG/2.2) [73] and concentration of total lipids (TL) was estimated using Phillips formula (TL = 2.27TCh + TG + 0.62) [74,75].

### 4.5. Additional Covariates in GESTE 

Additional data collected in the GESTE study were utilized to characterize the study sample and to address potential confounding by factors with effects on child lipid levels and correlated with in utero POPs. Questionnaires were administered by interviewers at the first prenatal visit and after delivery and were used to collect information on education, income, alcohol consumption, smoking and other factors [35]. Trained staff measured women’s height and weight at the enrolment visit and collected self-reported information on the usual weight and height after delivery. Children’s height and weight was similarly assessed at the age 6–7 years old. All data on maternal health and obstetrical history, medication use, and delivery were obtained from medical records.

In addition to questionnaire data, data for metals was considered as metal exposures have been shown in many studies to affect lipid metabolism (reviewed in [76]). Whole blood manganese, lead, cadmium and mercury were determined by the Toxicology Center of Quebec at the Quebec Institute for Public Health (CTQ-INSPQ) as described elsewhere [34,35]. In short, cold vapor atomic absorption spectrophotometry was used for total blood mercury and mass spectrometry plasma torch (ICP-MS) for lead, cadmium and manganese. Limits of detection were 2.07 μg/L for lead, 0.1 μg/L for mercury, 0.05 μg/L for cadmium, and 4.4 μg/L for manganese. 

### 4.6. Data Management and Statistical Analyses

Chemical levels below the LOD were set to LOD/2. In order to address outliers related to measurement error and undue leverage in analysis, values higher than 3.5 standard deviations above the mean were replaced by this cut point [77]. Some variables were natural log transformed based on the visual inspections of distributions to meet distributional assumptions for statistical modeling. 

Generalized linear models (GLM) were used to evaluate the relation of childhood lipid levels as continuous variables with in utero POPs. The interpretation of regression coefficients varies depending on whether a log transformation was used for independent and dependent variables. In recognition of the limited statistical power for this pilot study, a conservative approach to specifying multivariable models was used to build parsimonious models, aimed at maximizing statistical power while controlling for confounding. Using underlying biology and existing literature, a priori potential confounding factors based on plausible causal relations with child lipid levels and POP exposures was identified. These a priori potential confounders included: maternal age (years); smoking during pregnancy; alcohol using during pregnancy; child sex (male/female); maternal education; maternal body mass index at 1st trimester and after delivery (BMI, (weight (kg) /[height (m)]^2^)); child’s weight and BMI at age 6–7; delivery type (vaginal, caesarean, missing); gestational diabetes during current pregnancy; maternal delivery and 6–7 year children’s serum levels of lead, cadmium, manganese, and mercury. Only those factors related to both outcome and exposure (*p* < 0.20) were retained in final models to address confounding, as determined from ANOVA for categorical variables and generalized linear models for continuous variables. For the primary statistical models of lipids, chemical exposures were included as continuous variables. To evaluate departures from the assumption of linearity in the GLM models, chemical exposures were categorized into tertiles. Regression models of serum lipid outcomes were run as non-parametric assessments of dose–response patterns.

We utilized data on age 6–7 child chemical levels to account for potential confounding of in utero chemical effects by later life exposure. Using available data on chemical levels measured at age 6–7 (including BDE-100, BDE-99, BDE-153, and BDE-183), we evaluated associations with maternal delivery levels by correlation analysis, and associations with child lipid levels using GLMs. Finally, multivariable regression models were run including childhood POPs related to in utero exposures and child lipid outcomes as covariates in order to estimate effects of in utero exposure independent of later life chemical levels. 

In addition to the main analyses described above, we evaluated differences by offspring sex to address sex-differences in associations by performing stratified analyses as well as including a cross-product interaction term. This interaction term was used to test additive interaction between child sex and exposure. For all analyses, statistical significance was defined as *p*-values < 0.05.

## 5. Conclusions

In the current study, our data are consistent with long-term effects of in utero environmental exposures to PCB and PBDE on blood lipids, previously observed in murine models, but to this date, not assessed in humans. Relationships between higher in utero POPs and lower childhood blood lipids may indicate increased lipid accumulation in livers, a mechanism of NAFLD development. Additionally, we observed shift in children lipid profiles by ongoing exposures to several PBDE congeners, consisting in the increased LDL-C and decreased HDL-C, a potential risk factor for cardiovascular disease. Our study provides suggestion that long-term changes in blood lipids may be sexually dimorphic, in line with well-established sexual dimorphism in liver physiology. Given the modest sample size of our study and recognition that this sexual dimorphism and implications for lipid metabolism increase drastically after puberty, further research on bigger child cohorts as well as on adult subjects is needed to study long-term programing of liver lipid metabolism and sex-specific effects of developmental exposures to POPs.

## Figures and Tables

**Figure 1 metabolites-11-00657-f001:**
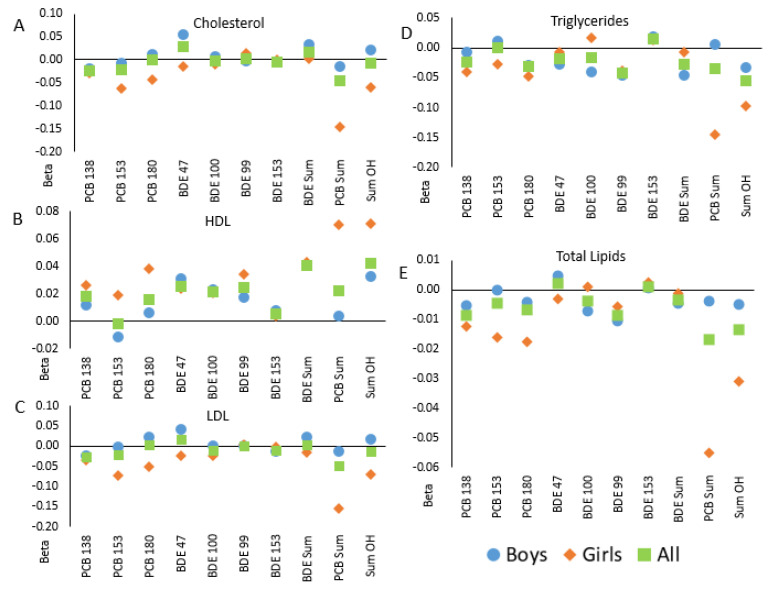
Correlations between in utero POP levels and blood lipids in children in the GESTE cohort, overall and separate by child sex: (**A**)—total cholesterol; (**B**)—high density lipoproteins; (**C**)—low density lipoproteins, (**D**)—triglycerides; (**E**)—total lipids. Linear regression coefficients are shown from models of lipids measured in children at age 6–7 from the GESTE cohort and in utero levels of PCBs and PBDEs. Squares indicate results in the full cohort (*n* = 147); offspring sex-specific estimates are shown with circles (in male offspring only, *n* = 82) and diamonds (female offspring only, *n* = 65).

**Table 1 metabolites-11-00657-t001:** Participant-Pairs Characteristics by Total Lipid Tertiles (mmol/L). GESTE Cohort.

	Tertile 1, *n* = 51 (Lower-Upper) ^+^		Tertile 2, *n* = 45 (Lower-Upper) ^+^		Tertile 3, *n* = 51 (Lower-Upper) ^+^		
	4.05	0.35	4.72	0.15	5.55	0.46	
	Mean *	(SD)	Mean *	(SD)	Mean *	(SD)	*p* ^#^
Maternal age (years) at pregnancy	30.3	(3.9)	30.0	(4.4)	27.8	(4.9)	0.009
Maternal BMI after delivery	26.7	(4.5)	26.6	(5.3)	27.4	(6.2)	0.76
Child BMI at age 6–7 years	15.8	(1.4)	15.9	(1.5)	15.9	(1.9)	0.95
PCB Sum (ng/mL)	0.090	(3.34)	0.086	(2.71)	0.066	(4.63)	0.43
BDE Sum (ng/mL)	0.135	(3.56)	0.123	(3.66)	0.155	(3.93)	0.68
Triglycerides (mmol/L)	0.697	(1.39)	0.794	(1.36)	1.393	(1.43)	<0.0001
Maternal serum lead levels at delivery (umol/L)	0.037	(1.46)	0.037	(1.62)	0.041	(1.71)	0.35
Maternal serum manganese levels at delivery (nmol/L)	247.0	(1.27)	253.3	(1.35)	282.0	(1.29)	0.03
Maternal serum cadmium levels at delivery (nmol/L)	2.008	(2.09)	1.716	(1.65)	2.117	(2.02)	0.29
Maternal serum mercury levels at delivery (μg/L)	2.585	(2.00)	2.740	(1.67)	2.351	(2.04)	0.51
Child serum lead levels at 6–7 years (umol/L)	0.033	(1.58)	0.037	(1.52)	0.035	(1.53)	0.54
Child serum manganese levels at 6–7 years (nmol/L)	172.2	(1.32)	185.6	(1.30)	178.8	(1.27)	0.37
Child serum cadmium levels at 6–7 years (nmol/L)	0.800	(1.41)	0.795	(1.31)	0.803	(1.27)	0.99
Child serum mercury levels at 6–7 years (μg/L)	1.569	(1.83)	1.474	(1.88)	1.512	(1.84)	0.88
	%		%		%		
% Female child	35.3%		48.9%		49.0%		0.28
% Reported smoking during pregnancy ^a^	6.0%		0.0%		7.8%		0.18
% Reported alcohol use during pregnancy ^a^	30.0%		25.0%		25.5%		0.83
% Vaginal delivery ^a^	82.4%		75.6%		84.3%		0.53
Maternal educational level							0.42
Primary	0.0%		0.0%		2.0%		
Secondary	9.8%		8.9%		19.6%		
College	19.6%		24.4%		29.4%		
University	35.3%		42.2%		27.5%		
Other	31.3%		22.2%		21.6%		
Missing	3.9%		2.2%		0.0%		

Abbreviations: SD-standard deviation, BMI-body mass index (kilograms/(meters2)), PCB-Polychlorinated biphenyl, BDE-Decabromodiphenyl ether. ^+^ Lower and upper bounds for the tertiles of total lipids, measured by Phillip’s Method, in mmol/L. * Mean and SD represent geometric mean and geometric SD. ^#^
*p*-Value from one-way ANOVA for all linear characteristics. For categorical characteristics, Chi-Squared tests were used except for smoking and maternal education level, which required Fisher’s Test. ^a^. These groups had 2 participants with missing information.

**Table 2 metabolites-11-00657-t002:** Distributions of organohalogens (ng/mL) among mothers within GESTE cohort.

	% below LOC	Minimum	25% Q1	50% Median	75% Q3	Maximum	Q3 ÷ Q1
BDE 47	0%	0.00035	0.020	0.05	0.11	1.05	5.6
BDE 99	13%	0.00005	0.004	0.02	0.06	0.64	14.1
BDE 100	12%	0.00005	0.010	0.02	0.06	0.43	5.8
BDE 153	16%	0.00005	0.008	0.03	0.06	0.87	8.3
Sum BDEs	--	0.00241	0.064	0.14	0.34	2.49	5.3
PCB 138	20%	0.00001	0.002	0.02	0.04	0.27	18.1
PCB 153	5%	0.00001	0.024	0.05	0.07	0.41	3.0
PCB 180	10%	0.00001	0.009	0.03	0.05	0.26	4.8
Sum PCBs	--	0.00003	0.039	0.09	0.16	0.91	4.1
Sum of All Organohalogens	--	0.00244	0.14	0.27	0.55	2.64	3.9

Total number of participants included in analyses was 147. Abbreviations: PCB-Polychlorinated biphenyl, BDE-Decabromodiphenyl ether.

**Table 3 metabolites-11-00657-t003:** Relationship between Chemical Exposures Levels at Delivery and Child Lipid Levels at 6–7 years in the GESTE Cohort.

		Bivariate Model			Multivariate Model ^+^		
	Chemical	Beta	SE	*p*	Beta	SE	*p*
Cholesterol ^a^	PCB 138	−0.028	(0.014)	0.04	−0.024	(0.014)	0.09
	PCB 153	−0.029	(0.022)	**	−0.022	(0.022)	**
	PCB 180	−0.003	(0.019)	**	0.001	(0.018)	**
	Sum PCB	−0.062	(0.037)	0.09	−0.046	(0.037)	**
	BDE 47	0.020	(0.032)	**	0.029	(0.031)	**
	BDE 100	−0.005	(0.020)	**	−0.004	(0.019)	**
	BDE 99	0.002	(0.018)	**	0.003	(0.018)	**
	BDE 153	−0.009	(0.017)	**	−0.006	(0.017)	**
	Sum BDE	0.011	(0.036)	**	0.015	(0.035)	**
	Sum OH	−0.021	(0.045)	**	−0.007	(0.044)	**
HDL ^b^	PCB 138	0.018	(0.010)	0.06	0.018	(0.010)	0.07
	PCB 153	−0.001	(0.016)	**	−0.002	(0.016)	**
	PCB 180	0.015	(0.013)	**	0.015	(0.013)	**
	Sum PCB	0.023	(0.026)	**	0.022	(0.027)	**
	BDE 47	0.026	(0.022)	**	0.025	(0.022)	**
	BDE 100	0.021	(0.013)	**	0.021	(0.014)	**
	BDE 99	0.024	(0.013)	0.06	0.024	(0.013)	0.06
	BDE 153	0.006	(0.012)	**	0.005	(0.012)	**
	Sum BDE	0.041	(0.025)	**	0.041	(0.025)	**
	Sum OH	0.043	(0.031)	**	0.042	(0.031)	**
LDL ^a^	PCB 138	−0.026	(0.014)	0.06	−0.022	(0.013)	**
	PCB 153	−0.026	(0.022)	**	−0.022	(0.022)	**
	PCB 180	0.011	(0.018)	**	0.014	(0.018)	**
	Sum PCB	−0.049	(0.036)	**	−0.037	(0.036)	**
	BDE 47	0.012	(0.031)	**	0.022	(0.030)	**
	BDE 100	−0.010	(0.019)	**	−0.007	(0.019)	**
	BDE 99	0.012	(0.018)	**	0.014	(0.018)	**
	BDE 153	−0.017	(0.017)	**	−0.013	(0.016)	**
	Sum BDE	0.005	(0.035)	**	0.011	(0.034)	**
	Sum OH	−0.008	(0.043)	**	0.004	(0.043)	**
Triglycerides ^b^	PCB 138	−0.026	(0.011)	0.02	−0.023	(0.011)	0.04
	PCB 153	−0.008	(0.018)	**	0.000	(0.018)	**
	PCB 180	−0.033	(0.015)	0.02	−0.032	(0.015)	0.03
	Sum PCB	−0.046	(0.029)	**	−0.035	(0.030)	**
	BDE 47	−0.015	(0.025)	**	−0.018	(0.025)	**
	BDE 100	−0.014	(0.016)	**	−0.016	(0.015)	**
	BDE 99	−0.041	(0.014)	0.01	−0.043	(0.014)	0.003
	BDE 153	0.016	(0.014)	**	0.015	(0.014)	**
	Sum BDE	−0.024	(0.029)	**	−0.028	(0.028)	**
	Sum OH	−0.060	(0.035)	0.09	−0.055	(0.035)	**
Total lipids ^b^	PCB 138	−0.010	(0.004)	0.01	−0.009	(0.004)	0.02
	PCB 153	−0.007	(0.006)	**	−0.004	(0.006)	**
	PCB 180	−0.008	(0.005)	**	−0.007	(0.005)	**
	Sum PCB	−0.022	(0.010)	0.02	−0.017	(0.010)	0.09
	BDE 47	0.001	(0.008)	**	0.002	(0.008)	**
	BDE 100	−0.004	(0.005)	**	−0.004	(0.005)	**
	BDE 99	−0.008	(0.005)	0.09	−0.009	(0.005)	0.07
	BDE 153	0.001	(0.005)	**	0.001	(0.005)	**
	Sum BDE	−0.004	(0.010)	**	−0.004	(0.009)	**
	Sum OH	−0.017	(0.012)	**	−0.013	(0.012)	**

Abbreviations: SE-standard error, P-T-test *p*-value, HDL-high density lipoproteins, LDL-low density lipoproteins, PCB-polychlorinated biphenyl, BDE-brominated diphenyl ether, OH-organohalogens. ^a^ Beta and SE from model using cholesterol untransformed and log transformed chemicals. ^b^ Beta and SE from model using log transformed exposure and log transformed outcome. ** *p*-value from T-test ≥ 0.10. ^+^ Adjusted for mother’s age during pregnancy and mother’s body mass index after delivery.

**Table 4 metabolites-11-00657-t004:** Interaction between Chemical Exposures Levels at Delivery and Child Biological Sex on Child Lipid Levels at 6–7 years in the GESTE Cohort.

		Bivariate Model	Multivariate Model ^+^
	Chemical	*p* *	*p* *
Cholesterol ^a^	PCB 138	0.66	0.75
	PCB 153	0.36	0.34
	PCB 180	0.34	0.22
	Sum PCB	0.16	0.14
	BDE 47	0.16	0.18
	BDE 100	0.47	0.55
	BDE 99	0.98	0.75
	BDE 153	0.94	0.99
	Sum BDE	0.48	0.52
	Sum OH	0.27	0.33
HDL ^b^	PCB 138	0.56	0.56
	PCB 153	0.45	0.46
	PCB 180	0.29	0.29
	Sum PCB	0.35	0.35
	BDE 47	0.85	0.85
	BDE 100	0.93	0.93
	BDE 99	0.48	0.50
	BDE 153	0.85	0.85
	Sum BDE	0.97	0.96
	Sum OH	0.62	0.63
LDL ^a^	PCB 138	0.69	0.77
	PCB 153	0.24	0.21
	PCB 180	0.16	**0.08**
	Sum PCB	0.15	0.13
	BDE 47	0.17	0.19
	BDE 100	0.34	0.41
	BDE 99	0.70	0.92
	BDE 153	0.91	0.85
	Sum BDE	0.42	0.46
	Sum OH	0.28	0.34
Triglycerides ^b^	PCB 138	0.12	0.16
	PCB 153	0.25	0.32
	PCB 180	0.46	0.56
	Sum PCB	**0.02**	**0.02**
	BDE 47	0.76	0.76
	BDE 100	**0.07**	**0.08**
	BDE 99	0.996	0.88
	BDE 153	0.86	0.80
	Sum BDE	0.50	0.55
	Sum OH	0.38	0.36
Total lipids ^b^	PCB 138	0.28	0.37
	PCB 153	0.26	0.28
	PCB 180	0.30	0.25
	Sum PCB	**0.03**	**0.03**
	BDE 47	0.47	0.51
	BDE 100	0.59	0.53
	BDE 99	0.95	0.70
	BDE 153	0.97	0.96
	Sum BDE	0.99	0.99
	Sum OH	0.23	0.27

Abbreviations: SE-standard error, P-T-test *p*-value, HDL-high density lipoproteins, LDL-low density lipoproteins, PCB-polychlorinated biphenyl, BDE-brominated diphenyl ether, OH-organohalogens. ^a^ Beta and SE from model using cholesterol untransformed and log transformed chemicals. ^b^ Beta and SE from model using log transformed exposure and log transformed outcome. * *p*-value from Type 3 F-test. ^+^ Adjusted for mother’s age during pregnancy and mother’s body mass index after delivery. Bold values indicate significant and semi-significant findings.

## Data Availability

Data are available on request to the PI of the GESTE cohort L.Takser due to protect human subjects.

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
