# Peer review of "In Utero Exposure to Persistent Organic Pollutants and Childhood Lipid Levels"

_metabolites, 2021, doi:10.3390/metabo11100657_

Round 1

Reviewer 1 Report

there is very specific and it will be interesting for this domain researchers

Author Response

We are grateful to the reviewer for the positive comments

Reviewer 2 Report

With the manuscript “In utero exposure to persistent organic pollutants and childhood lipid levels”, authors reported the influence exposure to POPs in fetal period on the blood lipid levels in 6-7 years-old. The results presented in this paper are novel and should be interested by global readers. the results are clear and convincing. To ensure the data interpretation and improve the discussion, I have one suggestion to be considered by the Authors.

  1. Discussion needs the more explanation of the molecular mechanism of POP-dependent altering the lipid levels and its sex differences. How does POP affect lipid metabolism? through CAR and PXR activation? Why POP-dependent alteration of lipid level was different depending on the type of POP? What cause the sex differences? Why sex difference was observed in the limited type of POPs? Please consider to describe.

Author Response

We are very thankful to the reviewer for the positive comments. We have added a paragraph in the discussion discussing the potential mechanistic link between POPs lipid metabolism and sex.

Reviewer 3 Report

Good and timely article on an important issue. The article is very well written and well structured. I support publication the current format.

Author Response

We are very thankful to the reviewer for the positive review.